# Personalized models of disorders of consciousness reveal complementary roles of connectivity and local parameters in diagnosis and prognosis

Lou Zonca[1]*, Anira Escrichs[1], Gustavo Patow[1,2], Dragana Manasova[3,4], Yonathan Sanz-Perl[1], Jitka Annen[5,6], Olivia Gosseries[5,6], Steven Laureys[5,7,8], Jacobo Diego Sitt[3], Gustavo Deco[1]

**1** Center for Brain and Cognition, University Pompeu Fabra, Barcelona, Spain, **2** Girona University, Girona, Spain, **3** Sorbonne Université, Institut du Cerveau - Paris Brain Institute - ICM, Inserm, CNRS, Paris, France, **4** Université Paris Cité, Paris, France, **5** Coma Science Group, GIGA-Consciousness, University of Liege, Liege, Belgium, **6** NeuroRehab & Consciousness Clinic, Neurology Department, University Hospital of Liège, Liège, Belgium, **7** Joint International Research Unit on Consciousness, CERVO Brain Research Centre, Laval University, Quebec City, Quebec, Canada, **8** International Consciousness Science Institute, Hangzhou Normal University, Hangzhou, China

* lou.zonca@upf.edu

**Data availability statement:** All the data necessary to reproduce the results from the

## Abstract

The study of disorders of consciousness (DoC) is very complex because patients suffer from a wide variety of lesions, affected brain mechanisms, different severity of symptoms, and are unable to communicate. Combining neuroimaging data and mathematical modeling can help us quantify and better describe some of these alterations. The goal of this study is to provide a new analysis and modeling pipeline for fMRI data leading to new diagnosis and prognosis biomarkers at the individual patient level. To do so, we project patients' fMRI data into a low-dimension *latent-space*. We define the latent space's dimension as the smallest dimension able to maintain the complexity, non-linearities, and information carried by the data, according to different criteria that we detail in the first part. This dimensionality reduction procedure then allows us to build biologically inspired latent whole-brain models that can be calibrated at the single-patient level. In particular, we propose a new model inspired by the regulation of neuronal activity by astrocytes in the brain. This modeling procedure leads to two types of model-based biomarkers (MBBs) that provide novel insight at different levels: (1) the connectivity matrices bring us information about the severity of the patient's diagnosis, and, (2) the local node parameters correlate to the patient's etiology, age and prognosis. Altogether, this study offers a new data processing framework for resting-state fMRI which provides crucial information regarding DoC patients diagnosis and prognosis. Finally, this analysis pipeline could be applied to other neurological conditions.

paper can be downloaded from Zenodo: https://doi.org/10.5281/zenodo.15847853 All the codes from the manuscript are available at https://github.com/louzonca/DoC_models.

**Funding:** LZ was supported by the FLAG-ERA JTC2021 project ModelDXConsciousness (Human Brain Project Partnering Project) Grant PCI2021-122019-2A funded by MICIU/AEI /10.13039/501100011033 and by the European Union NextGenerationEU/PRTR. AE was supported by the project eBRAIN-Health—Actionable Multilevel Health Data (id 101058516), funded by EU Horizon Europe and by the Grant PID2022-136216NBI00, funded by MICIU/AEI/10.13039/501100011033, and "ERDF A way of making Europe", ERDF, EU. YSP was supported by the project NEurological MEchanismS of Injury, and the project Sleep-like cellular dynamics (NEMESIS) (ref. 101071900) funded by the EU ERC Synergy Horizon Europe. GD was supported by the project NEurological MEchanismS of Injury, and the project Sleep-like cellular dynamics (NEMESIS) (ref. 101071900) funded by the EU ERC Synergy Horizon Europe and and by the Grant PID2022-136216NB-I00, funded by MICIU/AEI/10.13039/501100011033, and "ERDF A way of making Europe", ERDF, EU. This research was partially funded by Grant PID2021-122136OB-C22 funded by MICIU/AEI/10.13039/501100011033 and by ERDF A way of making Europe for GP. DM received individual funding from Ecole Doctorale Frontières de l'Innovation en Recherche et Education–Programme Bettencourt. DM and JDS were supported by funding from the EU ERAPerMed Joint Translational Call for Proposals for "Personalised Medicine: Multidisciplinary research towards implementation" (ERA PerMed JTC2019). JDS was also supported by the FLAG-ERA JTC2021 project ModelDXConsciousness (Human Brain Project Partnering Project). The study was further supported by the University and University Hospital of Liège, the BIAL Foundation, the Belgian National Funds for Scientific Research (FRS-FNRS), the FNRS PDR project (T.0134.21), the FLAG-ERA JTC2021 project ModelDXConsciousness (Human Brain Project Partnering Project) and FLAG-ERA JTC 2023 - HBP - Basic and Applied Research, project BrainAct, JTC the fund Generet, the King Baudouin Foundation, the Funds Chantal Schaeck Yolande, the Télévie Foundation, the European Space Agency (ESA) and the

## Introduction

The question of consciousness, or lack thereof, is a complicated and controversial notion that has attracted the attention of theologists [1], philosophers [2], psychologists, and medical doctors [3] for centuries and remains under active investigation. In recent years, scientists and researchers have been investigating novel ways to characterize and quantify different states of consciousness. In particular, disorders of consciousness (DoC) refer to a scope of conditions that can occur after patients who suffered various types of brain damage (e.g., traumatic brain injury (TBI), stroke, or cardiac arrest leading to anoxia) fall into a coma from which they only partially recover. There are two main dimensions used to characterize DoC patients: (1) *arousal*, which refers to wakefulness or vigilance, and (2) *awareness*, of the environment or the self [4]. Based on these two dimensions, medical doctors have defined several categories of DoC patients: the Vegetative State or Unresponsive Wakefulness Syndrome (VS/UWS) [5] corresponds to patients who do not show any conscious responses to sensory stimulation; the Minimally Conscious State (MCS) [6] describes patients who show limited but clear evidence of awareness. MCS has been further divided into MCS– and MCS+ [7] based on the level of language understanding. However, each of these categories includes very heterogeneous populations of patients in terms of pathology, behavior, and cognitive ability. The introduction and evolution of neuroimaging and electrophysiology allowed researchers to better quantify several aspects of DoC patients' responses, specifically the notion of covert consciousness, leading to a refined description of consciousness. This led to the discovery and definition of Cognitive-Motor Dissociation (CMD) [8], which describes patients who exhibit clear cognitive abilities despite the absence of motor responses. Using these techniques, consciousness has been theorized to be supported by cortical and subcortical awareness networks, involving multiple regions of the brain [4,9–12]. This led to the development of many quantitative scores or empirical biomarkers (EBMs) based on neuroimaging and/or electrophysiology data. Among them, the Functional Connectivity matrix (FC), which can be extracted from fMRI data at resting state (rs-fMRI), was proven to be sufficient to distinguish between UWS and MCS [13]. Furthermore, previous misdiagnosis of CMD patients [14] is a striking advocate for the development of accurate biomarkers. Indeed, a recent study showed that at least 25% of unresponsive patients could be classified as CMD patients [15]. Moreover, the chances of recovery for a CMD patient are significantly higher than for UWS patients [16] and a too early removal of life-support for such patients can have dramatic consequences.

However, although EBMs are theoretically driven, they remain purely descriptive of the patient's state and do not bring proper information about the underlying brain functions that are disrupted and therefore, cannot be used to investigate therapeutic approaches, which remain scarce today [11,17–19]. In this context, the development of whole-brain computational models emerged in parallel with the evolution of EBMs to bring a complementary understanding of the affected mechanisms in DoC patients. Indeed, for clinicians who aim to improve a patient's condition, it is fundamental to mechanistically understand what is missing in the patient's brain function to design a specific intervention that could restore the lost cognitive ability. These models are based on the fact that the dynamical co-activations and correlations between different small brain regions are shaped by the underlying structural connectivity (SC) [20–22]. Several studies have shown the crucial role of the SC and FC matrices in classifying patients into different groups representative of the patient's state using both fMRI [23,24] and MEG [25] data during resting state. A correct classification of a patient's state is crucial for prognosis and decision-making, e.g., regarding the maintenance of life-support.

Belgian Federal Science Policy Office (BELSPO) in the framework of the PRODEX Programme, the Public Utility Foundation 'Université Européenne du Travail', "Fondazione Europea di Ricerca Biomedica", the BIAL Foundation, the Mind Science Foundation, the European Commission, the Fondation Leon Fredericq, the Mind-Care foundation, the National Natural Science Foundation of China (Joint Research Project 81471100) and the European Foundation of Biomedical Research FERB Onlus, the Horizon 2020 MSCA – Research and Innovation Staff Exchange DoC-Box project (HORIZON-MSCA-2022-SE-01-01; 101131344). OG is Research Associate at FRS-FNRS. JA is postdoctoral fellow funded (1265522N) by the Fund for Scientific Research-Flanders (FWO). The funders had no role in study design, data collection and analysis, decision to publish, or preparation of the manuscript.

**Competing interests:** The authors have declared that no competing interests exist.

However, existing methods describe generic states in which patients can be categorized, but they drastically lack individuality as well as biological interpretability. Without individualization, there cannot be a systematic investigation of the underlying mechanistic or biological disruptions responsible for the patient's loss of consciousness, as they may be different from one patient to the next. Furthermore, these models solely explore the meso-scale inter-regional connectivity aspects underlying brain function, and none account for specific, local, biological processes that could play an important role in consciousness-related mechanisms. In particular, the regulation of neuronal activity by the astrocyte network could play a critical role in the diagnosis or recovery of some patients. Indeed, it is now well known that astrocytes play a crucial role in neuronal activity [26–32], which is also reflected in the BOLD signal [33] due to their strategic role in neurovascular coupling [34–36]. The main astrocyte pathways are metabolism regulation (through the astrocyte-neuron lactate shuttle), glutamate and GABA regulation, and extracellular potassium. Recent studies have shown the particular importance of the potassium pathway on neuronal bursting activity [37–39]. Other crucial reasons why we think that the regulatory role of astrocytes should be considered, in addition to its fundamental role in brain activity, are, first, that in severe brain lesions, astrocytes probably have been as impaired as neurons. Indeed, astrocytes are present in all brain regions and are even believed to be the most numerous cell type in the brain [40,41]. Second, an increasing body of literature points to the fundamental role of astrocyte regulation of sleep, both in rapid eye motion (REM) and slow wave sleep [42–45], which points to their active role in healthy consciousness mechanisms. Therefore, a solely neuron-centric approach seems limited if we want to understand the mechanisms underlying DoC, for at least some of our patients.

In this paper, we propose to investigate DoC patients, using biologically interpretable brain models that we will calibrate to each patient to grasp the differences between them. However, this approach requires working in very high-dimensional spaces, which leads to a prohibitive number of model parameters to fit and heavy computations. Furthermore, a large body of recent studies [46–53] have unveiled the existence of a low-dimensional space that can encompass the information contained in the observed brain resting-state activity such as fMRI recordings. This reduced space must still account for the nonlinearities inherent to the neuronal activity that contains all the complexity underlying resting state brain dynamics and is commonly called the *latent space* or *latent manifold*.

In the present work, we started by investigating the optimal dimension to take the best advantage of our data without over-complexifying our models. In particular, we combined different dimensionality reduction methods (namely, auto-encoders and principal component analysis) to find the optimal (in a sense that we will define) latent space to project our data. Once the data was projected in the latent space, we built *latent* whole-brain models that we calibrated to each patient individually. Specifically, we compared a classical Hopf model with a new, biologically inspired model that accounts for astrocyte regulation of neuronal activity through the potassium pathway described above. Finally, we used the fitted model parameters as model-based biomarkers (MBBs), which we analyzed at two different scales: The fitted connectivity matrices, called the generative effective connectivity matrices (GEC), showed that they carry the information about the severity of the patient's diagnosis. We then studied the connection between latent dimensions and the resting-state networks [54] in the natural space. Finally, we took a closer look at the local node parameters of the models and revealed that they carry information related to the patient's etiology, age and prognosis.

## Results

### A low-dimensional latent space preserving information

The main difficulty when fitting a model to data is having sufficient information to constrain the values of the, sometimes high, number of free parameters. Fitting meso-scale brain models, ranging from around 100 to 1000 nodes (depending on the initial atlas chosen) at the single-patient level thus constitutes a challenge. Indeed, the amount of information available in one fMRI recording of 5 min (approximately 300 time-points in our case), although informative, is limited, compared to the more classical approach of fitting one model for a group of dozens of subjects suffering the same condition. To overcome this limitation, we decided to explore data dimensionality reduction methods that would project the high-dimensional fMRI data (with as many dimensions as the parcellation size) into a lower dimensional space, also called the latent space. The reasons for doing this are multiple: first, our group recently showed that projecting the data in a reduced latent-space reduced the redundancies in the data, in addition to reducing the number of model parameters needed, thus allowing a more precise fit of the models [51]. Second, this approach also discards difficult open questions regarding fMRI preprocessing. Indeed, fMRI pre-processing is subject to various decisions on which there is still no consensus. These decisions include the choice of the parcellation atlas and whether to denoise or regress out several components of the signal (e.g., global signal regression). In this context, dimensionality reduction can be used to avoid these choices as we will look for the minimal dimension that preserves the information present in our data, thus eliminating any noise or redundancies in the signal.

In particular, we explored two open questions: (1) does the initial parcellation influence the dimension of the latent space and what is this *optimal* dimension? (2) Does the dimensionality reduction method matter?

**The initial parcellation does not influence the dimension of the latent space.** To determine the optimal dimension of the latent space, we trained Auto-Encoders (AE, Fig 1A, purple) with different bottleneck dimensions ranging from 2 to 25. Briefly, an AE is an artificial neural network composed of successive layers of reducing dimension (the encoder) until reaching the bottleneck, or latent dimension, and a symmetric group of layers of increasing dimension (the decoder) that brings the data back to its original dimension. The AE is then trained to minimize the difference between input and output. The optimal latent space dimension is considered to be where the curve of the mean squared reconstruction error (MSE) vs latent dimension presents an elbow: indeed, when the latent dimension is too small, there is a loss of information and thus the reconstruction is bad, but then, increasing the latent dimension reduces the error until reaching the actual dimension needed to carry all the information contained in the data and, from this point forward, adding more dimensions does not improve the reconstruction (see methods Sect Auto-encoders for more details).

Here, to determine whether the original parcellation influenced the results, we performed this analysis on the same data (see methods Sect Participants for description of the dataset) that we parcellated using the Schaefer parcellation [55] with 100 regions (Fig 1B light purple), then, we added the subcortical regions using the Tian parcellation [56] with 16 regions (pink), and finally we tried the Schaefer parcellation with 1000 regions (deep purple). We found no difference in the MSE between the different parcellations. Furthermore, the elbow was reached in all cases around dimension 15. Therefore, for the rest of the results of this study, we used the Schaefer 100 parcellation. This was done using different datasets, from two different centers (in Paris and Liege) of fMRI data from DoC patients with UWS and MCS diagnosis and healthy controls (see Methods Sects Participants and Datasets combination for details).

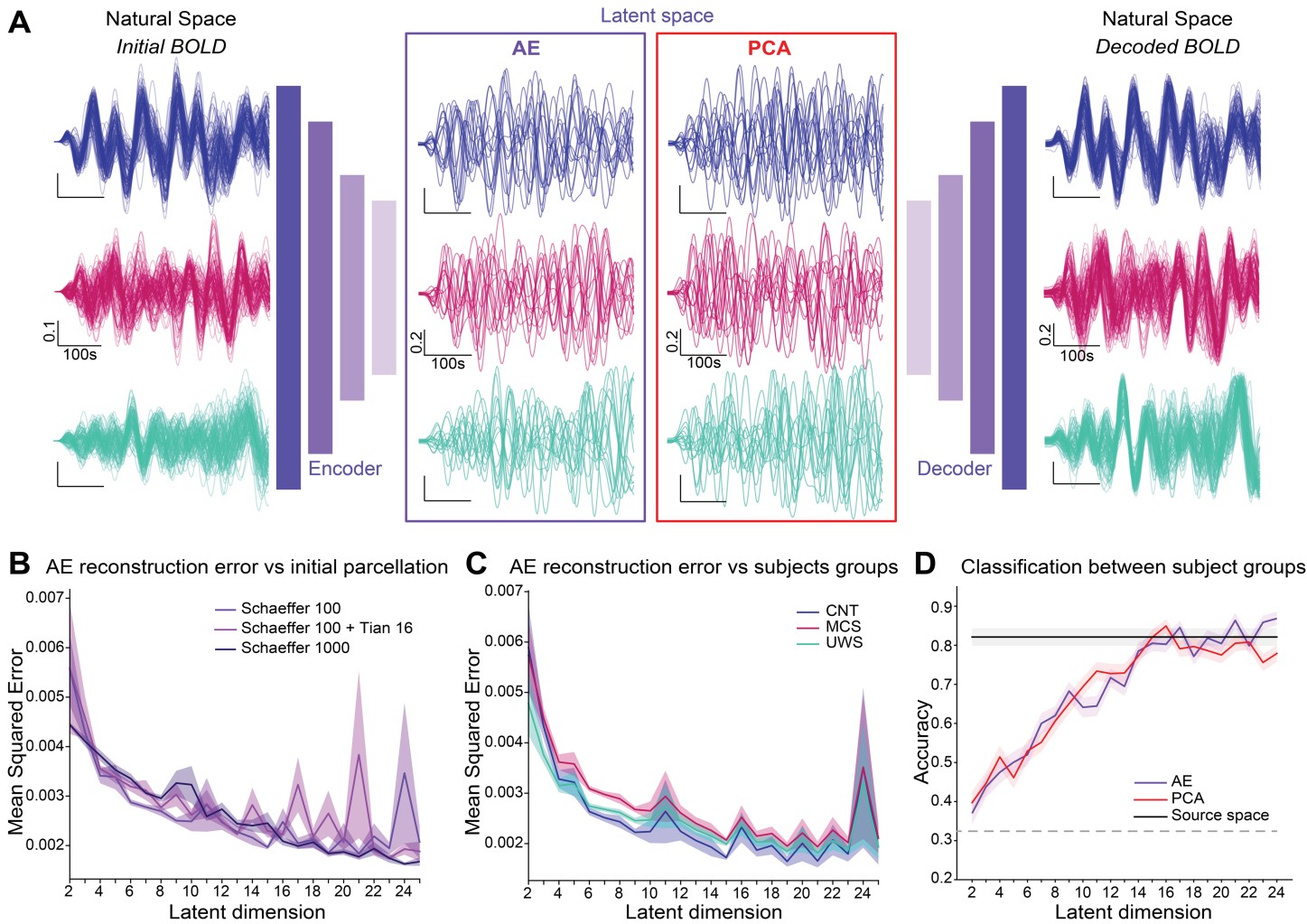

**Fig 1. Latent space exploration.** A: Example subjects' initial rs-BOLD time-series (left) color-coded per condition (blue: CNT, magenta: MCS, cyan: UWS) were projected into a reduced latent space (center) either using an AE (center left, purple) or PCA (center right, red) and then decoded (for the AE) trying to minimize the reconstruction error (right). B: MSE Reconstruction error for the AE with respect to the latent space dimension for different initial parcellations with and without sub-cortical regions (shades of purple). C: MSE vs latent dimension for initial parcellation Schaefer 100, divided per subjects condition. D: Comparison of the classification accuracy between conditions for AE (purple) vs PCA (red) for all latent dimensions, compared to accuracy in the natural space (black solid line). Grey dotted line indicates chance level.

We also checked whether the patient's diagnosis would influence the elbow's position by using the previously trained AEs (trained on a balanced mix of subjects from all diagnosis and healthy controls (CNT)). We then split the subjects of the test set by condition (i.e. patients' diagnosis: UWS or MCS, or healthy controls) and looked at the MSE by group vs the latent-dimension (Fig 1C control (CNT), blue, MCS, magenta, UWS, cyan), and found that there was no difference in the optimal latent-dimension based on the subjects' condition.

**The latent dimension minimizes the MSE and allows the classification between conditions better than in the natural space.** To make sure that the dimensionality reduction did not lead to a loss of information, we decided to use a second criterion: given that we already knew that, in the natural space we can distinguish between clinical conditions based on the FC matrices of the subjects [13], we wanted to make sure that, in the latent space we were

still able to do it, at least as well. To do so, we trained a support vector machine (SVM) classifier in the source space (of dimension 100) to distinguish between the empirical FC of all our subjects that are grouped into three categories based on the clinical exams: UWS, MCS, and CNT. We could obtain a very good accuracy of over 82%. We then used a similar training procedure on the reduced data for all latent dimensions between 2 and 25 and for the two dimensionality-reduction methods (PCA and AE). The training of the SVMs in the reduced dimensions was done on the latent empirical FCs, i.e. the Pearson correlation matrices of the reduced signals, i.e. for each reduced dimension $n \in [[2, 25]]$ we obtained latent FC matrices of dimension $n \times n$ on which we trained the SVM classifier. The rationale was that the latent dimension should be the smallest one that allows for similar, or even better, classification accuracy than in the source space. We found that dimension $d_{opt}$ = 15 was the optimum for both methods. The fact that both methods gave the same results is a good sign of robustness, showing that the intrinsic level of information contained in the data is of this optimal dimension, regardless of the dimensionality reduction method used.

To conclude, we found that both AE and PCA give similar results regarding the classification of the FC matrices between clinical conditions. However, PCA is a linear process while we know that the processes underlying the generation of the fMRI signals that we observe are intrinsically non-linear. For this reason we decided to pursue our study using the AE. Finally, we projected all the available fMRI data into this latent space of optimal dimension 15, to start working on the modeling step that we describe in the next section.

## Modeling rs-fMRI activity in the latent space

In this section, we introduce two models that we used to deepen our understanding of the data. Our goal here was to use the Hopf model [57] (see Sect Hopf model) as a *control* model, in the sense that this model has already proven very valuable for fMRI analysis, and then to push forward our analysis by introducing a novel model, that we shall describe below.

As mentioned, we implemented two different models: (a) the Hopf model, and (b) a new model based on synaptic short-term plasticity that accounts for the regulatory effect of astrocytes on neuronal activity. This is done through the modulation of a neuronal regulatory mechanism called afterhyperpolarization (AHP) [37,58]. This last model simulates the neuronal response, which we then combined with the Balloon-Windkessel model [59,60] to simulate the final BOLD response. We decided to add this new model (that we refer to as the AHP model in what follows) to our study to explore whether this regulatory effect is affected depending on the conditions because of the known impact of astrocytes in the regulation of sleep and brain oscillations [30,42]. Indeed, this model allows us to have a direct biological interpretation of the fitted parameters. In other words, it provides model-based biomarkers (MBBs) with a direct biological meaning.

**Individual patient fitting procedure.** For both models, there are two types of parameters: (i) the local, or node parameters, e.g., the bifurcation parameter (often denoted *a*) for the Hopf model, which determines the type of activity (random fluctuations or synchronized oscillations); or the parameters relative to AHP in the case of the second model. (ii) The connectivity matrix between the nodes, also known as the Effective Connectivity (EC) matrix, which gives the actual level of influence each node of the model has on the remaining ones.

We implemented the model fitting procedure (Fig 2A) as follows: (1) We first fitted the node parameters by minimizing the distance between the empirical and simulated FC, i.e., the $15 \times 15$ Pearson correlation matrices between the time series of the 15 nodes. We then computed the structural similarity index (SSIM) [61] between the empirical and simulated FC,

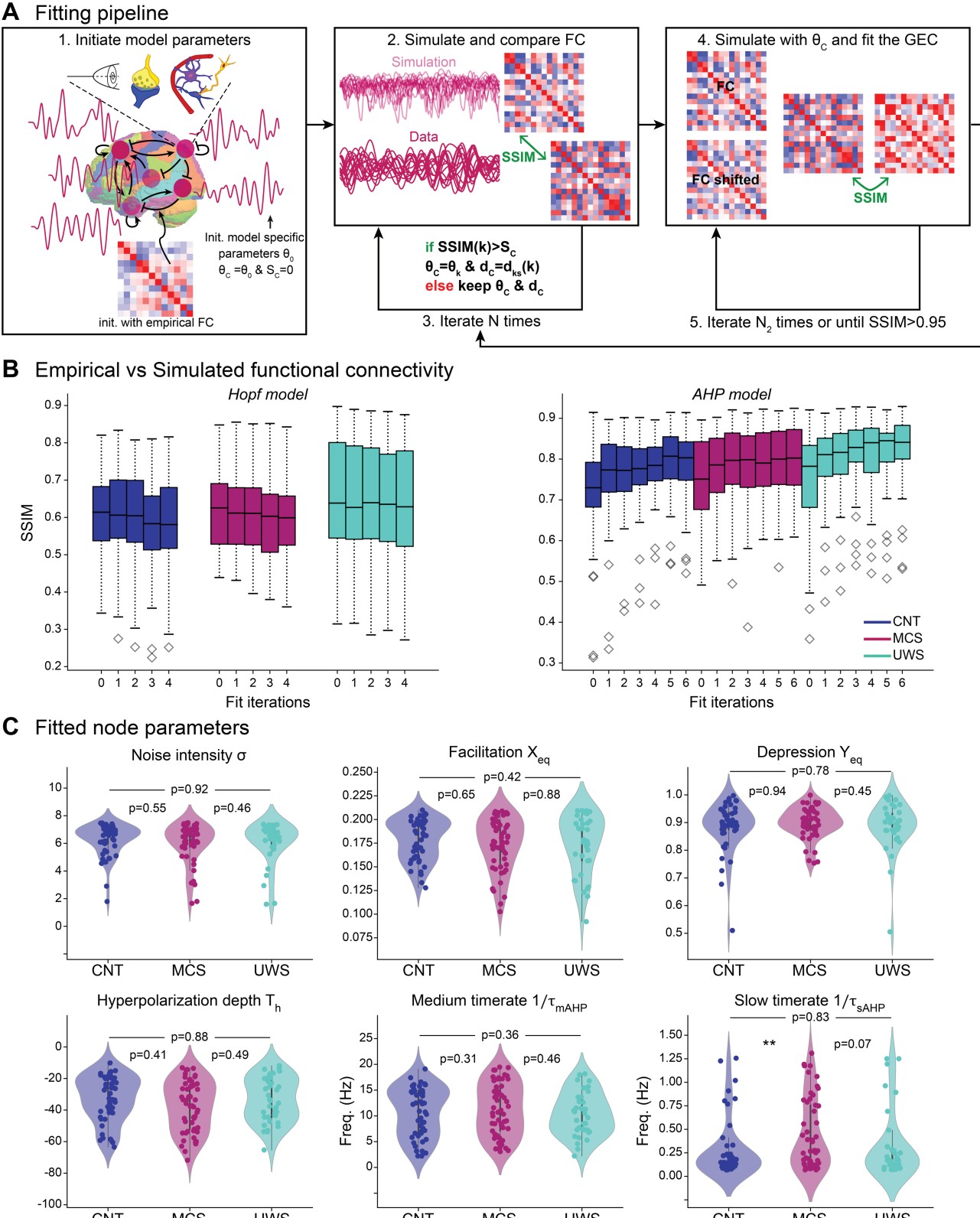

**Fig 2. Models fitting.** A: Iterative fitting procedure: the node parameters were randomly initialized and the connectivity was initialized by the empirical FC (step 1, left). Then, the subject's latent time series were simulated and the empirical and simulated FC were compared to maximize the SSIM (step 2, center) over the fitting iterations with random exploration of the parameter space (step 3). Then, with the best node parameters fixed, the GEC was fitted iteratively to maximize the SSIM between simulated and empirical FC and shifted FC (steps 4 and 5, right). The procedure was iterated at step 2,

to fit the node parameters using the fitted GEC. B: Fitting results: SSIM between the simulated and empirical FC after various fitting iterations per condition (blue: CNT, magenta: MCS, cyan: UWS) for the Hopt (left) and AHP (right) models. C: 1st level MBBs per condition, obtained from the fitted node parameters (each dot is one subject, top left to bottom right: noise intensity $\sigma$, facilitation resting-state value $X_{eq}$, depression resting-state value $Y_{eq}$, hyperpolarization depth $T_h$, all dimensionless, medium $1/\tau_{mAHP}$ and slow $1/\tau_{sAHP}$ AHP time rates (in Hz), each dot is one patient.

which we maximized as a fitting criterion (a SSIM of 1 means perfect similarity and 0 none). Indeed, previous studies have shown that the SSIM offers a good trade-off between absolute (e.g MSE) and relative (e.g. correlation) assessment of the differences between two matrices [62]. The parameter space was then sampled randomly until a threshold $SSIM \geq 0.9$ was reached, or until the maximum number of simulations $N_{max}$ was reached. We kept the parameter set that maximized the SSIM. (2) The second step of the procedure was to iteratively fit the generative effective connectivity (GEC) for each patient by comparing the empirical FC to the simulated one in a similar manner as described above, using the previously fitted node parameters (see Methods, Sect Fitting algorithm, for more details).

We found that the fit quality plateaued around 0.6 for the Hopf model (Fig 2B, left) and was constant over iterations. Note that in [62], a SSIM of 0.4 is already considered as a good fit, here, we considerably increased it. In contrast, the SSIM of the AHP model (Fig 2B, right) increased over the first iterations before stabilizing around 0.80-0.86 (depending on the group, the UWS had the best-fit quality, although the difference was small).

Since we have more node parameters (6 vs 2) in the case of the AHP model compared to the Hopf model, it is of no surprise that the fit would be of better quality. Therefore, in the following sections, we present the results obtained from this model. However, all analyses were performed on both models, with the results for the Hopf model being available in the supplementary information (SI). We used this double analysis as a control to see whether we get coherent results with both models to make sure that we did not see model artifacts but robust results. In brief, the AHP model provided us with a more precise fit and interpretation than the Hopf model, but both models point to similar results, as we shall see in the next sections.

**Node fitting results: Model-based biomarkers.** Each patient's set of fitted node parameters constitutes model-based biomarkers (MBB, Fig 2C). For the AHP model, we obtained a vector of dimension 6 for each patient (the fitted node parameters are, from top left to bottom right: the noise intensity $\sigma$, the facilitation resting state $X_{eq}$, the depression resting state $Y_{eq}$, and the three AHP parameters: hyperpolarization depth $T_h$, and medium and slow timerates $1/\tau_{mAHP}$ and $1/\tau_{sAHP}$). We first checked whether these parameters revealed differences between the three groups (CNT in blue, MCS in magenta, and UWS in cyan) but, apart from the slow AHP time rate $1/\tau_{sAHP}$ that was significantly increased (i.e. indicating a shorter AHP duration) in the MCS group compared to the controls, as well as a close to significant difference (p=0.07) between MCS and UWS, the other parameters didn't show any clear difference based on the clinical state. We found the same results for the Hopf model MBBs, which are only 2 (noise intensity $\sigma$ and the bifurcation parameter $a$, see S1 Fig). However, each distribution was quite spread out thus revealing a wide within-group variability. At this point, we concluded that the node MBBs do not carry much information about the clinical state, and therefore do not allow to distinguish between conditions but that they may hold other type of information about the patients' conditions such as their etiology, age, underlying affected mechanisms, and other clinical variables, as we explore in the last section of this study. But first, we still needed to recover and explain the difference between the conditions we knew we could distinguish from (Fig 1D).

## Analysis of the GEC reveals the differences between conditions

**A two-clique structure for the GEC.** To analyze the difference between the groups we thus turned to the GEC matrices. Fig 3A shows the averaged fitted GEC matrix per condition. However, this visualization makes it hard to see the differences between them. Thus, to further explore the structure of these GEC matrices, we converted each GEC matrix into a weighted graph, where the GEC is the connectivity matrix of the graph. Then, for a clearer visualization, we split these graphs into two subgraphs: one with positive edges (Fig 3B, upper line) and another one with negative edges (lower line). The first striking observation was that, for all groups, these graphs are constituted of two groups of nodes (or cliques), strongly positively connected within each group and negatively projecting to the other clique. At first glance, we seemed to have a stronger within-clique positive density for the CNT (0.66), than for the MCS (0.65) and lowest for the UWS (0.60), while we had the opposite trend for the negative inter-clique projections (CNT: 0.49, MCS: 0.50 and UWS: 0.54).

We thus decided to further quantify the clique interactions (Fig 3C). To do so, we used the *networkx* package to automatically compute the two major cliques based on the selection of each node, and computed the mean and standard deviation of the inter-clique connectivity for the positive (Fig 3C, left), and negative subgraphs (right). We found that these measures, not only allowed us to differentiate the controls from the patients, but also between the two patients groups with high significance.

We also looked at other graph measurements (Fig 3D), namely the mean node centrality for the positive subgraph (first left), which displayed a small difference (p=0.07) between CNT and MCS but not between the others; and the mean node centrality of the negative subgraph (second left), which allowed to distinguish between CNT and MCS but not between the others. The small-world clustering (Fig 3D, left) for the positive edges showed a significant difference between CNT and MCS and a fairly significant difference (p=0.08) between CNT and UWS, but failed to distinguish between MCS and UWS, and the same for the negative edges, where it could only differentiate between CNT and UWS.

Finally we also directly quantified values from the GEC matrices (Fig 3E): from left to right: the sparsity (quantified as the proportion of connections <0.015), the asymmetry (difference between the GEC matrix and its transpose), and the mean connectivity and its standard deviation (i.e., the mean and std values of the matrix components). Similarly, we found no clear difference between the two DoC groups. However, the mean connectivity seemed higher in both DoC groups than for the CNT, indicating a more synchronized activity, as expected. We saw the same trends using the Hopf model (see Fig S2).

In conclusion, it seems that the difference between conditions is mostly carried by this two-clique structure we observe in the GEC graphs, with stronger negative inter-clique and weaker positive inter-clique projections in the least conscious patients than in the more conscious ones and the controls.

**Decoding the latent dimensions correlates to the resting state networks.** To further understand the meaning of the two-clique structure observed in the GEC matrices (Fig 3B), we decoded the signals from the latent GEC cliques and each latent dimension to see how they correlate to the natural structures in the brain. In particular, we investigated the link between GEC cliques (resp. latent dimensions) and the 7 Yeo Resting State Networks (RSN) [54], with a similar approach to the one used in [51] (see Methods, Sect Decoding of the GEC cliques and latent dimensions).

Briefly, we decoded signals with activity (from the control subjects) on only one of the two GEC cliques. We computed the resulting FC matrix (Fig 4A, left, and 4B) for the average over all control subjects. We then compared these two FC matrices with those

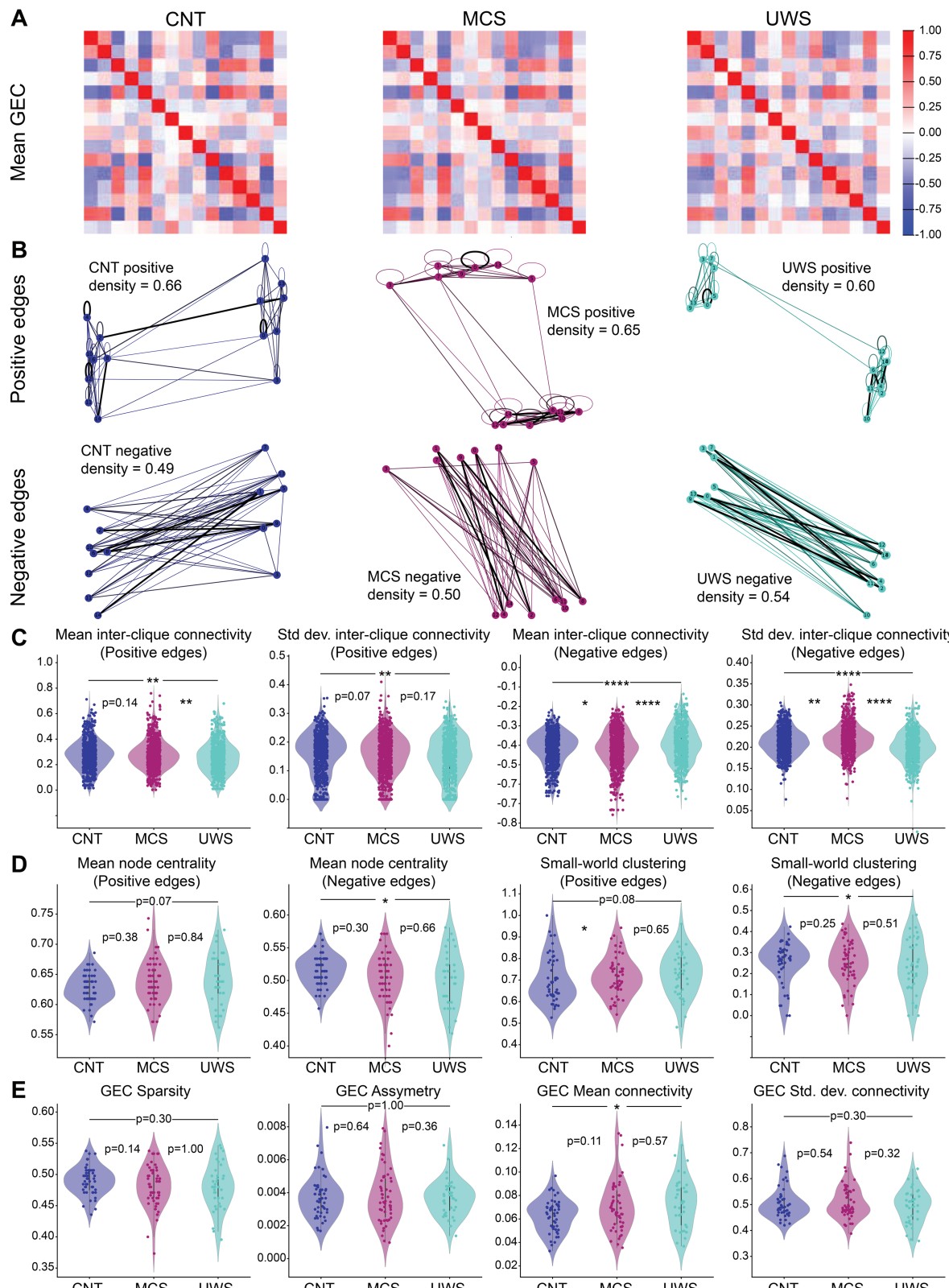

**Fig 3. GEC-based biomarkers for the AHP model.** A: Average over all subjects of the fitted GEC matrix per condition (left to right, CNT, MCS, and UWS). B: Extracted graphs from the average GEC matrices (top: positive connections, bottom: negative connections) showing

two subgroups (or cliques) of nodes positively connected between them and projecting negatively on the other subgroup, for each condition. C: Inter-clique metrics: mean and standard deviation for positive (left) and negative (right) edges. D: Graph metrics: node centrality and Small-world clustering parameter. E. GEC extracted measures: left to right: sparsity, asymmetry, mean and standard deviation values. Statistical tests: pairwise Kolmogorov-Smirnov tests, ****: $p \leq 0.0001$, ***:$0.0001 < p \leq 0.001$, **:$0.001 < p \leq 0.01$, *:$0.01 < p \leq 0.05$.

corresponding to activity on each of the 7 RSNs (Fig 4A, right). For both cliques, we computed the SSIM between the clique FC and each RSN FC (Fig 4C). We found that the visual network (light gray) projected most strongly on Clique 0, followed by the Dorsal Attention network (blue). In contrast the default mode network (DMN, beige) and the Ventral Attention network (purple) were more strongly associated with Clique 1.

We then decided to investigate the connection between each latent dimension $LD \in [[0, 14]]$ and the 7 RSNs. Using a similar approach, we compared the FC matrices from decoded signals with activity on only one of the $LD$ at a time to the RSN FC matrices (Fig 4D). We found that the Somatomotor network (light blue), was most strongly associated with latent dimensions $LD13$ and $2$, while the DMN strongly projected onto $LD1$ and $9$. We also noted that $LD11$ was mainly related to the Ventral attention network. Generally, most networks were associated with a combination of latent dimensions rather than one in particular.

This could mean that the lower within-clique density observed in UWS patients is related to lower or less consistent activity on specific RSNs. However, a more detailed analysis would be required to confirm such a hypothesis.

## The node MBBs reveal transverse clusters that correlate to etiology and outcome

In this section, we used a subset of our patients for whom we had more detailed diagnosis (splitting MCS between MCS+, and MCS–) as well as detailed metadata, including age, gender, etiology and evolution of the condition (different Coma Recovery Scale-Revised (CRS-R) diagnosis at different points in time after the patient's admission) as well as the outcome (see methods Sect Paris subset used for the cluster analysis).

As we pointed out before, DoC patients are a very heterogeneous group in symptoms and causes. Thus, we decided to explore the parameter space (the virtual patients or node MBBs) while being agnostic on the diagnosis. The first step was to look for a low-dimensional representation of the parameter space that could allow us to visualize the emergence of transverse clusters of patients for both models. We did so by using Uniform Manifold Approximation and Projection for Dimension Reduction (UMAP) [63] (Fig S3). We then recovered the emerging clusters that we saw in Fig S3A using the *HDBScan* clustering algorithm [64] (Fig S3B, 5A and S4A).

We could then see which parameters carried the specific distinction between clusters (Fig 5B). For example, separation into cluster 0 (red) is mainly carried by the facilitation resting state parameter $X_{eq}$ (Fig 5B top, center, red), as well as the hyperpolarization depth $T_h$ (bottom left), while the distinction between clusters 3 (light green) and 4 (purple) seems driven by a combination of all three AHP parameters (bottom row).

To understand the interpretation of these clusters, we checked how belonging to a specific cluster correlated with the available metadata (Fig 5C). We plotted the age distribution of subjects of each cluster (Fig 5C top left), and the gender balance (top right), neither of which exhibited clear tendencies. Then the etiology (anoxia, traumatic brain injury (TBI), combined anoxia and TBI, stroke, SAH, and all others, Fig 5C bottom left) and the outcome (see

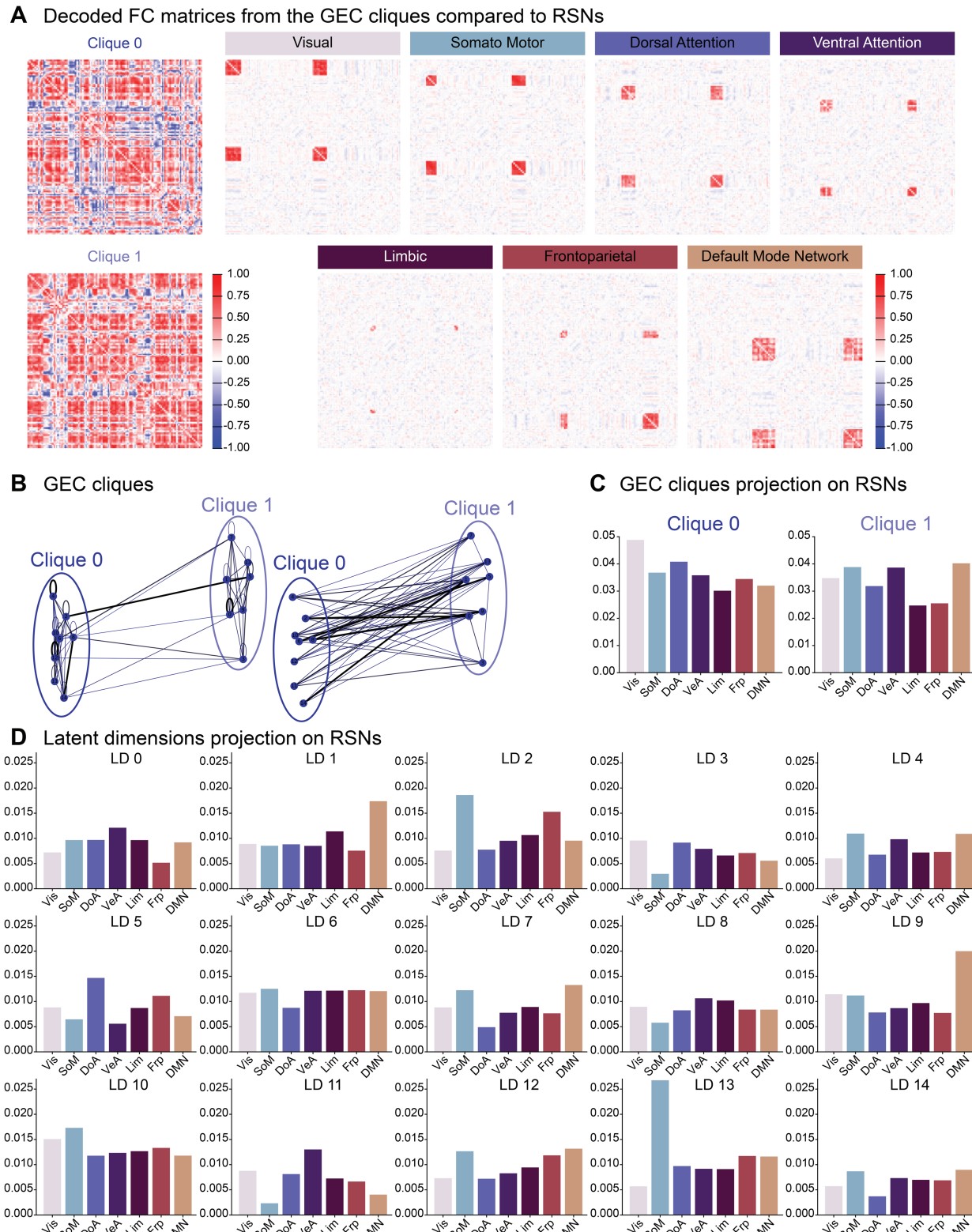

**Fig 4. GEC cliques and latent dimensions connection to the Resting State Networks.** A: FC matrices from the decoded signals for both GEC cliques (left), and FC matrices corresponding to the 7 RSNs in the natural space. B: GEC cliques from the average CNT GEC graphs. C: SSIM between the decoded FC matrices from each GEC clique compared to each RSN. D: Same as C but for the decoded signals from each latent dimension.

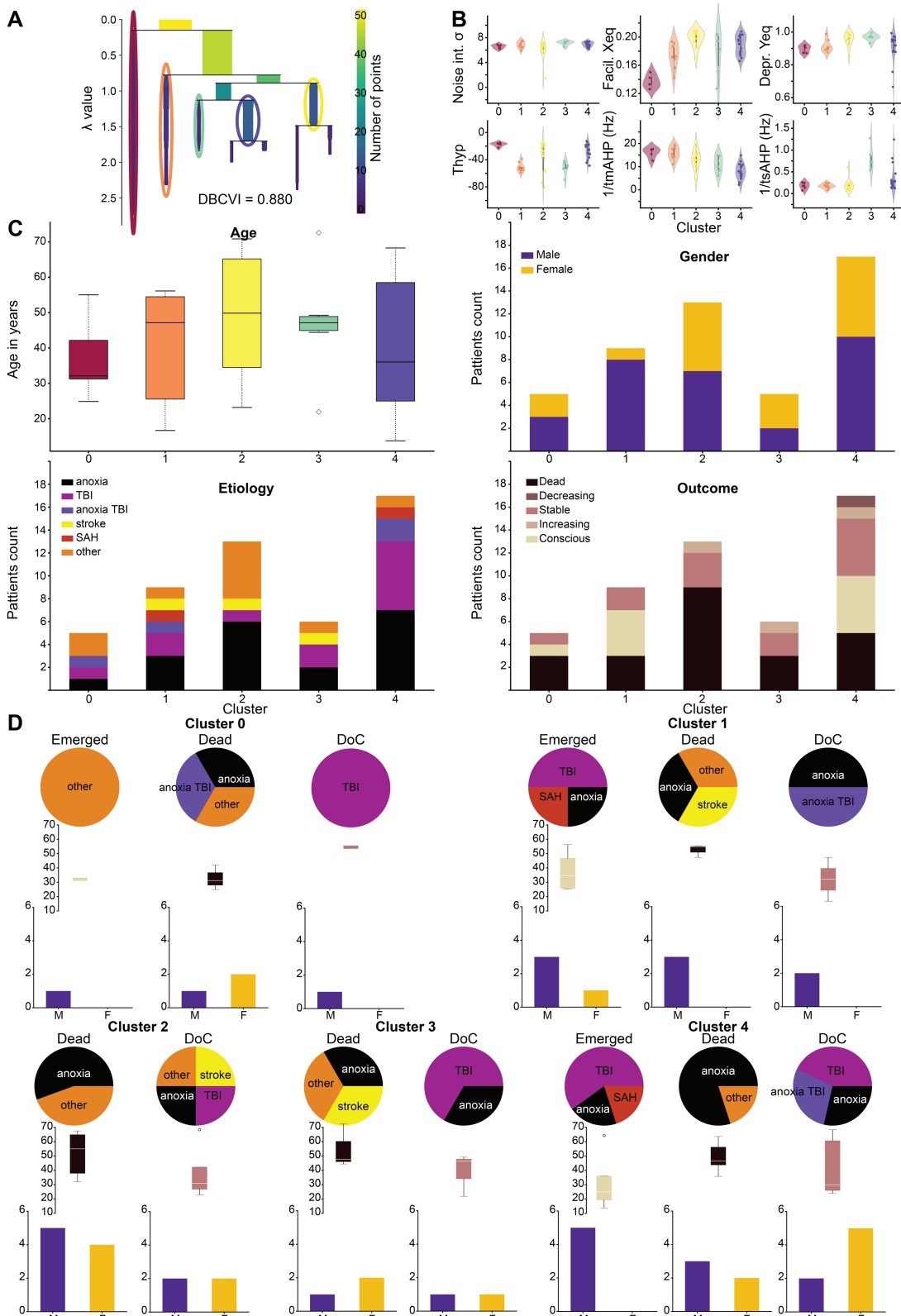

**Fig 5. Transverse clusters from the AHP model node parameters.** A: Condensed tree plot of the *HDBSCAN* clustering algorithm showing the level of similarity between clusters. B: Fitted node parameters, separated by cluster, showing which

parameters influence the attribution to which clusters. C: Composition of the clusters: top left, the age distribution in years of the patients belonging to each cluster. Top right, gender distribution within clusters. Bottom left, etiology, bottom right, outcome. D: Co-factor analysis within clusters: for each cluster, we split the patients between emerged, still in DoC, and dead, and show the distribution of their etiology, age, and gender balance.

Methods Sect Outcome categorization), which we split into dead, decreasing (i.e., when the evolution of the CRS-R diagnosis went towards less conscious over checks but did not lead to death), stable, increasing (when the diagnosis ameliorated without leading to consciousness), and conscious/emerged (Fig 5C bottom right). Here we saw some very interesting tendencies: for example, clusters 2 and 3 contained mainly patients with negative outcomes, with a majority of deaths (respectively: 66.7% and 60%) and some stable DoC patients (33.3% and 40%) and 0 recovery in either cluster, while half of cluster 1 consists of emerged patients (50%). Finally, clusters 0 and 4 contained more mixed outcomes including dead (resp. 50% and 31.25%), emerged (25% each), and DoC patients (resp. 25% and 43.75%), as summarized in Table 1.

Since it is known that those factors (in particular, age, etiology, and outcome) are not independent (for example, we know that anoxia patients have a poorer prognosis), we further analyzed the composition of the clusters (Fig 5D): for each cluster, we split the patients between emerged, dead and DoC (grouping, increasing, stable and decreasing patients) and then looked at the distribution of etiology, age and gender with each category. Here, we see that for the clusters with mainly negative outcomes (clusters 2 and 3) both age and etiology correlate to the prognosis, with all the dead being older with 53, resp. 55, years old on average and who mainly suffered from anoxia (5 of the 9 deceased patients in cluster 2), while the stable DoC patients were the younger ones (39 years old on average in both clusters) and all of the ones with TBI associated to these clusters. However, compared to other TBI DoC patients who were associated with clusters with more favorable outcomes, this could be interpreted as a potentially worse prognosis since their MBBs were closer to those of people with a negative prognosis. In the same way, when we looked at cluster 1, which contains mainly favorable outcomes we saw that all the TBI patients from this cluster emerged, as well as some anoxic and the SAH ones. In the meantime, we saw once again that the few dead patients in this cluster (3 subjects) were the oldest ones (52 years old on average). When we looked into the mixed outcome clusters (0 and 4), similarly, we saw that age is a good predictor of the outcome since the emerged ones were the youngest patients (32, years old on average in both clusters), while the dead ones were almost all anoxic and older patients (33, resp. 49 years old). Finally, we note that gender did not seem to exhibit any clear tendencies regarding its correlation to one or the other of the observed co-factors (see Table 1).

Going back to the distribution of the node MBBs by cluster (Fig 5B), we could see that, for the negative outcome clusters 2 and 3 (yellow and light green) facilitation was quite high (similarly as in cluster 4, but higher than in the good outcome cluster 1). Facilitation can be

**Table 1. Summary of clusters composition.**

|  | Emerged | Dead | DoC | M | F | Age | TBI | Anoxia | Other |
|---|---|---|---|---|---|---|---|---|---|
| Cluster 0 | 25% | 50% | 25% | 25% | 75% | 37.08 ± 10.55 | 25% | 25% | 50% |
| Cluster 1 | 50% | 37.5% | 12.5% | 87.5% | 12.5% | 41.23 ± 14.05 | 25% | 37.5% | 37.5% |
| Cluster 2 | 0% | 66.7% | 33.3% | 50% | 50% | 48.48 ± 16.09 | 8.3% | 41.7% | 50% |
| Cluster 3 | 0% | 60% | 40% | 60% | 40% | 47.08 ± 14.70 | 20% | 40% | 20% |
| Cluster 4 | 25% | 31.25% | 43.75% | 56.25% | 43.15% | 41.06 ± 17.60 | 31.25% | 56.25% | 12.5% |

associated with pre-synaptic calcium concentration, this difference could thus hint at a deregulated calcium concentration. Still, for clusters 2 and 3 we saw that depression was close to one, which can be interpreted as unaltered compared to healthy people (a depression resting state parameter of 1 means that the readily releasable pool of pre-synaptic vesicles is filled at rest). However, we could see that cluster 3 (green) showed a higher slow AHP time rate, which corresponds to a shorter AHP period. In previous studies [37,58] we have associated this reduction in AHP with a disrupted astrocyte network and reduced astrocyte regulation of extracellular potassium levels. Similarly, we saw that cluster 2 seemed to be associated with a less deep hyperpolarization (parameter $T_h$). On the other hand, we have seen that facilitation (associated with presynaptic calcium) was very low in cluster 0 and the middle in cluster 1, while the distribution for cluster 4 seemed bimodal, with one-half at the level of cluster 1 and the other half closer to the values for clusters 2 and 3. Finally, cluster 0 seemed to have also a lower depression parameter, indicating more depressed synapses than in the healthy case. Finally, we summarized the results of the clusters composition in Table 1.

In conclusion, we saw that these novel node MBBs bring us two new types of information: (1) based on these MBBs, we can group the patients, regardless of their current diagnosis, into clusters that are correlated with the outcome and etiology, thus helping for a future, more precise diagnosis and prognosis. (2) The direct biological interpretation of these parameters can help us investigate the underlying affected mechanisms and form new hypotheses regarding the patient's condition and their potential treatment.

## Discussion

**Latent space dynamics and structural interpretation.** A growing body of literature shows evidence of a low-dimensional manifold that encompasses the brain's resting-state activity. The dimension of this latent space is always found to be of the same order of magnitude: around 10 to 15, across various studies, both from our group and from others working on different datasets and conditions [48–53]. Here, using rs-fMRI recording from DoC patients, we confirmed that finding and showed that we could work in a latent space of dimension 15 without losing the information contained in our data, or even our ability to explain it. This latent space description is not only convenient regarding the actual working dimension for explaining the data, but it also allows us to avoid inherent difficulties linked to fMRI preprocessing, parcellation choices, and denoising, as we showed in Fig 1. This reveals that the intrinsic amount of information contained in rs-fMRI data, from two different datasets, can be explained in such tractable dimensions, regardless of the condition studied.

However, the question of the biological interpretation of the latent space remains open. We believe that it should be interpreted in terms of interacting networks. In a recent study, we showed how the dimensions of the latent space (also called latent modes) project onto the resting state networks (RSNs) [51]. Here, we deepened this analysis of the connection between the latent dimensions and the RSNs (Fig 4). We found that each RSN seemed to be encoded by a combination of latent-dimensions, confirming that the results observed in the latent space have a biological interpretation. However, given the complexity and the amount of possible combinations, further research is still needed to decipher fully the mapping between the latent and natural spaces. Meanwhile, other works are unveiling the tight relation between spatial and temporal activity in the brain, or in other terms, how structure and function are intrinsically linked. For example, using harmonic decomposition, Atasoy and colleagues showed in several studies how the RSNs overlapped with functional task-based networks [65], thus revealing harmonic brain modes that could be compared to the latent modes. This

framework recently allowed us to reveal and quantify the effect of hallucinogenic drugs such as DMT [66] and ketamine [67]. In a similar attempt to connect the latent functional modes to the brain's structure, we recently showed that it was possible to extract the low-dimensional manifold information corresponding to the structural networks using Schroedinger's equation [68]. Similarly, Dan and colleagues developed deep-learning methods to link structural and functional brain dynamics [69,70]. Altogether, these results show that latent space dimensions are directly connected to structural brain networks, and thus ensure the biological interpretability of our latent whole-brain models, although more work is needed to get a more direct mapping between the two. Further developments in this direction shall shed even more light on our model interpretation shortly.

**Latent, biologically relevant, whole-brain models.** We implemented two different models with different levels of readability: the Hopf and the AHP models. This choice was motivated by the fact that we wanted to make sure that our results would not be model artifacts. This was confirmed given that they are consistent regardless of the model chosen at the node level. However, we also showed that using the AHP model leads to a better-fit quality (Fig 2B) while offering a straightforward biological interpretation for each of the fitted parameters (see Figs 2C and 5B, and the last paragraph of Sect The node MBBs reveal transverse clusters that correlate to etiology and outcome). In particular, this new model is paving the way for the development of neuron-glia whole-brain models. Indeed, in the past decades, it has become evident that neuron-only approaches were not sufficient to grasp and explain the complexities of the brain as well as to investigate its disorders. For example, recent modeling efforts have shown that astrocyte calcium activity could explain the neurovascular coupling leading to the BOLD response [33]. Initiating a direction of biologically motivated neuron-glia models *at the whole-brain level* is, to the best of our knowledge, still a niche approach. However, regarding the importance of glial cell regulation of brain activity, it seems fundamental. In this work, we showed that adding basic astrocyte regulatory mechanisms into our model already could increase our model quality and the data's explainability. In the future, it would be interesting to push this approach forward and work on implementing models that account more explicitly for astrocytes, and other types of glial cells, such as oligodendrocytes and microglia [71,72].

Finally, we point out that the fMRI processing pipeline provided here could also be used to study different datasets or conditions. Moreover, the modular structure of the provided code and latent whole brain models could even allow one to plug in any other model at the node level that would account for the mechanism one is interested in studying, without having to change the rest of the pipeline. We thus propose here a versatile analysis toolkit that could have multiple applications.

**The GEC carries the information on the condition.** Functional connectivity matrices were already known to be able to distinguish between UWS and MCS [13], and later works on effective connectivity confirmed these findings and showed that even better distinction could be made, in particular in the latent space [51]. The effective connectivity framework also allows us to distinguish between tasks such as rest vs movie watching [73]. In this study, we fitted Generative Effective Connectivity matrices, in a latent space, to analyze how they carried information about the severity of the DoC condition (Fig 3). In particular, using graph theory, we showed that our subjects' GEC presented two cliques of nodes, strongly connected positively but negatively projecting on the other subgroup (Fig 3B). Although this structure was similar in different conditions, we found significant differences between the three groups of subjects that we had. This analysis is thus a new step towards understanding in more detail the meso-scale dynamics that are underlying consciousness, and

how the GEC matrices carry them. This approach can be related to the circuit models of consciousness [10–12,74].

**Local node information correlates to prognosis and patient's etiology.** In the last part of this paper, we showed that looking into the local node parameters (node MBBs) revealed novel information about our patients' conditions (Fig 5). In particular, we found that these node MBBs, allowed us to group the patients into transverse clusters that are agnostic about the severity of the condition but are correlated to the etiology and evolution of the patients' condition, thus paving the way for novel prognosis tools or biomarkers. Moreover, the cluster separability analysis revealed which parameters were leading to belong to each cluster. Given the direct biological interpretation of the AHP model's parameters, this provides a new hypothesis regarding which mechanism is affected in which group of patients.

Finally, this work opens many possibilities for the investigation of potential treatments. Indeed, it would be very interesting to use the fitted models to see how modifying one specific node parameter affects the cluster to which a patient belongs. For example, if, by affecting one specific parameter (i.e., mechanisms or pharmacological pathway), we can move one patient from a least favorable outcome cluster to a more favorable one, this could be very informative about the treatments to explore for this specific patient. This type of analysis could thus lead to a more precision medicine approach to DoC treatments. Indeed, although many studies exist regarding treatment options both through external neuronal stimulations [19] and pharmacological approaches [11], the wide variability of lesions, and underlying causes for the DoC patients remains a barrier to generalization of such treatments.

## Materials and methods

### Ethics statement

We comply with all relevant EU legislation such as: The Charter of Fundamental Rights of the EU and notably with the Regulation (EU) 2016/679 of the European Parliament and of the Council of 27 April 2016 (General Data Protection Regulation). The present project follows the recommendations of the Declaration of Helsinki – Ethical Principles for Medical Research involving Human Subjects.

Human data is systematically anonymized. Written informed consent to participate in the study was obtained directly from healthy control participants and the legal surrogates of the patients.

Each partner providing data ensured that their work complies fully with all applicable local, government and international laws, regulations and guidelines, including those governing health and safety, data protection, and the use of human subjects and good clinical practice (GCP) including Belgium national legislation implementing the Parliament's Directive 2001/20/ECon GCP, the French national legislation implementing the Parliament's Directive 2001/20/EC on GCP, and the Parliament's Directive 2010/63/EU on the protection of animals used for scientific purposes. We are compliant with the ethical and scientific quality standards reported by the GCP guidelines of the European Union (Guideline for GCP E6(R1)) and 'GCP Directive' 2005/28/EC. All research activities comply with the ethical principles of Horizon 2020, French, Spanish and Belgian ethical regulations, European Union and international legislation, including the Charter of Fundamental Rights of the European Union, the European Convention on Human Rights and its Supplementary Protocols.

The datasets used here are the same ones that we used in previous studies and are fully described in [75,76].

## Participants

**Paris.** The dataset included 77 brain injury patients hospitalized at Paris Pitié-Salpêtrière. The patients' states of consciousness were evaluated using the Coma Recovery Scale-Revised (CRS-R) by trained clinicians. Patients were classified as being in an unresponsive wakefulness state (UWS) if they demonstrated arousal (such as eye-opening) but lacked any signs of awareness, such as voluntary movement. In contrast, those diagnosed with a minimally conscious state (MCS) exhibited behaviours suggestive of awareness, including visual tracking, response to pain, or consistent command-following. Exclusions from the study were made for individuals with errors in T1 image acquisition (n = 5), significant motion artefacts (n = 7), registration errors (n = 4), or large focal brain lesions (n = 4). As a result, the final cohort comprised 33 patients with MCS (11 females, mean age ± SD: 47.25±20.76 years), 24 patients with UWS (10 females, mean age ± SD: 39.25±16.30 years), and 13 healthy controls (7 females, mean age ± SD: 42.54±13.64 years). This research was approved by the local ethics committee, Comité de Protection des Personnes Ile de France 1 (Paris, France), under the protocol code 'Recherche en soins courants' (NEURODOC protocol, n 2013-A01385-40). Informed consent was obtained from the patient's families, and all procedures were conducted under the Declaration of Helsinki and French regulations.

**Liege.** The dataset involved 35 healthy controls (14 females, mean age±SD: 40±14 years) and 48 patients with DoC. Diagnoses were established following at least five Coma CRS-R assessments conducted by trained clinicians. The final diagnosis, confirmed using Positron Emission Tomography (PET), was determined by the highest level of consciousness recorded. Patients diagnosed with a minimally conscious state (MCS) showed relatively preserved metabolism in the fronto-parietal network, whereas those in an unresponsive wakefulness state (UWS) exhibited bilateral hypometabolism in the same network. The cohort ultimately included 33 patients in MCS (9 females, mean age±SD: 45±16 years) and 15 patients in UWS (6 females, mean age±SD: 47±16 years). The Ethics Committee of the Faculty of Medicine at the University of Liege approved the study protocol, and the research was conducted in compliance with the Helsinki Declaration. Written informed consent was obtained from controls and the legal representatives of the patients.

## MRI data acquisition

**Paris.** MRI images were obtained using two distinct acquisition protocols. The first protocol involved 26 patients and 13 healthy controls, with data collected on a 3T General Electric Signa System. Resting-state T2*-weighted images of the entire brain were acquired using a gradient-echo EPI sequence in an axial orientation (200 volumes, 48 slices, slice thickness: 3 mm, TR/TE: 2400 ms/30 ms, voxel size: 3.4375 × 3.4375 × 3.4375 mm, flip angle: 90°, FOV: 220 mm²). Additionally, an anatomical T1-weighted MPRAGE sequence was acquired during the same session (154 slices, slice thickness: 1.2 mm, TR/TE: 7.112 ms/3.084 ms, voxel size: 1 × 1 × 1 mm, flip angle: 15°). The second protocol involved MRI data acquisition from 51 patients on a 3T Siemens Skyra System. In this case, T2*-weighted whole-brain resting-state images were recorded with a gradient-echo EPI sequence in an axial orientation (180 volumes, 62 slices, slice thickness: 2.5 mm, TR/TE: 2000 ms/30 ms, voxel size: 2 × 2 × 2 mm, flip angle: 90°, FOV: 240 mm², multiband factor: 2). An anatomical volume was also acquired during the same session using a T1-weighted MPRAGE sequence (208 slices, slice thickness: 1.2 mm, TR/TE: 1800 ms/2.35 ms, voxel size: 0.85 × 0.85 × 0.85 mm, flip angle: 8°).

**Liege.** MRI images were obtained using a Siemens 3T Trio scanner (Siemens Inc, Munich, Germany). The acquisition protocol included a gradient echo-planar imaging (EPI) sequence with 32 transverse slices and 300 volumes (TR/TE: 2000 ms/30 ms, flip angle: 78°, voxel size:

$3 \times 3 \times 3$ mm, FOV: 192 mm). A structural T1-weighted scan was also acquired with 120 transverse slices (TR: 2300 ms, voxel size: $1.0 \times 1.0 \times 1.2$ mm, flip angle: 9°, FOV: 256 mm).

## Data pre-processing

The resting-state fMRI data was pre-processed using FSL (http://fsl.fmrib.ox.ac.uk/fsl), following the methodology outlined in our previous works [75,76]. Briefly, the analysis utilized MELODIC (Multivariate Exploratory Linear Optimised Decomposition into Independent Components) [77]. The steps involved discarding the initial five volumes, applying motion correction with MCFLIRT [78], performing brain extraction using BET (Brain Extraction Tool) [79], and spatially smoothing the data with a 5mm FWHM Gaussian kernel. Additional processing steps included rigid-body registration, applying a high-pass filter with a cut-off of 100.0 seconds, and conducting single-session independent component analysis (ICA) with automatic dimensionality estimation. Lesion-related artefacts (in patients) and noise components were removed for each subject using FIX (FMRIB's ICA-based X-noisier) [80]. Finally, the images were co-registered, and FSL tools were used to extract time-series data for each subject, which were mapped to MNI space using the Schaefer [55] and Tian [56] parcellations.

## Paris subset used for the cluster analysis

The preprocessing of the functional MRI data was performed using fMRIPrep (version 22.0.2) [81]. For anatomical T1-weighted scans that displayed visible lesions, lesion masks were created and incorporated into the fMRIPrep pipeline. This preprocessing included a 24-parameter head-motion correction [82], removal of the first four volumes, as well as low-pass and high-pass filtering (0.1 - 0.01 Hz). A total of 26 scans were excluded from the fMRI dataset due to preprocessing failures, such as misalignment or segmentation errors, often caused by patient movement or the size of the lesions. After applying lenient exclusion criteria, scans with a minimum of 180 volumes and a mean framewise displacement of less than 0.55 mm were retained for analysis.

## Latent space exploration

**Datasets combination.** In all that follows we combined the datasets from Paris and Liege described above in Sects Participants to Data pre-processing.

To overcome the problem of different time resolutions between the different datasets, we performed a simple time-interpolation step on the subsets with larger *TR* to obtain time-series that all have the same *TR* = 2s. This was done using the *interp*1*d* function from the *scipy.interpolate* package.

The decision of combining the different datasets was taken for the sake of robustness. Indeed, any classification algorithm trained on only one dataset would likely pick up some artifacts or features specific to the scanner used. This could result in overfitting to this specific dataset. Mixing different datasets allows us to avoid this pitfall and ensure the generalizability of our methods, regardless of the center and technical setup where the data was recorded. Furthermore, the preprocessing was strictly the same for all recordings from both centers, thus ensuring no differentiation between the datasets at this stage. Moreover, combining the datasets greatly increases the number of patients in each group and thus the statistical power and significance of any group-related analysis. We recall that the group classification (between MCS and UWS) used in the manuscript comes from the clinical diagnosis performed by the expert clinicians in Paris and Liege based on the CRS-R method which is a

universal framework for clinical diagnosis developed for that purpose to ensure that diagnosis of DoC patients is consistent between different centers. Finally, the dimensionality reduction also reduces possible differences due to the differences in data recording.

**Auto-encoders.** We used auto-encoders with three dense (i.e., fully connected) layers of dimension $0.75 \times d_{init}$, $0.5 \times d_{init}$ and $0.25 \times d_{init}$ before the bottleneck layer, where $d_{init} \in \{100, 116, 1000\}$ is the dimension of the natural space data depending on the initial parcellation (Schaeffer 100 and 1000 [55] and Schaeffer 100 with Tian 16 for the subcortical regions [56]). For each dimension of the bottleneck layer $d_{latent} \in [[2, 25]]$, we split the subjects into a 90-10% training-test split, then each data point given as input to the AE is a vector of dimension $d_{init}$ which corresponds to one time-point in the data time series of one patient. The mean squared reconstruction error (MSE) is then calculated as the MSE between the input and output vectors on the test set. For each $d_{latent}$ we performed a k-fold cross-validation using the *Kfold* function of the Python library *Sckit-learn* (*sklearn* v1.2.2) with $k = 10$.

The auto-encoders were implemented using the *Keras* library of *Tensorflow* v2.11 and trained using *cuda* 11.8 for GPU processing. Training hyperparameters are $LEARNING\_RATE = 0.0001$, $BATCH\_SIZE = 20$, $EPOCHS = 15$.

**Principal component analysis.** The PCA was implemented in Python using native Python functions and *Numpy*.

**SVM classifier.** We used the SVC method from the *sklearn* Python library v1.2.2 and leave-one-out cross-validation using the KFold method from *sklearn*. We used a degree 3 polynomial kernel with hyperparameters $C = 0.1$ and $\gamma = 7.5$. The algorithm was trained to distinguish the empirical Functional Connectivity matrices (FC) of the subjects. To enhance the classification accuracy, due to the limited number of subjects available, we previously split the data time-series into $n_{splits} = 4$ and then computed the FC of the split time-series which led us to $N = n_{splits} \times N_{sub} = 556$ FC matrices to train the SVM classifier, and 4 for the test (we do the Leave-one-out split on the subjects to avoid overfitting by training on some FC matrices of a subject and testing on the same subject).

## Models

**Hopf model.** The classical Hopf model is composed, in each node of the model, of the normal form of a supercritical Hopf bifurcation (Stuart-Landau oscillator), close to its bifurcation point. Here we have $n_{nodes} = d_{opt} = 15$ nodes in the latent space. The nodes are then connected via a connectivity matrix $C$, which we describe in Sect Fitting algorithm. The model equations at node $i$ are given by:

$$\dot{x} = (a - x_i^2 - y_i^2)x_i - \omega_i y_i + G \sum_{j=1}^{d_{opt}} C_{ij}\left(x_j - x_i\right) + \sigma\eta_i(t)$$

$$(1)$$

$$\dot{y} = (a - x_i^2 - y_i^2)y_i + \omega_i x_i + G \sum_{j=1}^{d_{opt}} C_{ij}\left(y_j - y_i\right) + \sigma\eta_i(t)$$

where $x$ is the real part of the system and $y$ its imaginary part, $G$ denotes the global coupling of the system, that we keep set to $G = 1$, $C$ is the connectivity matrix, $\omega_i$ is the intrinsic frequency at node $i$ (here we take the Spectral Edge Frequency at 50% (SEF50) value of the spectrogram of the signal) and $\eta_i$ is an additive Gaussian noise term at each node. We recall that $d_{opt} = 15$ is the dimension of the latent space and thus corresponds to the number of nodes in our model. Finally, $a$, the bifurcation parameter, and $\sigma$ the noise intensity are the parameters that we will fit for each patient (see Sect Fitting algorithm).

We chose to keep the global coupling parameter set to $G = 1$ because we fitted the entire connectivity matrix $C$ (Sect Fitting algorithm); thus, any changes in the global coupling intensity are integrated in the fitting of the connectivity matrix. From a mathematical point of view, $G$ being a constant it can be included in the sum and this is equivalent to fitting a connectivity matrix $J_{ij} = G \times C_{ij}$.

## AHP model

**Local AHP model.** For the AHP model we implemented a whole-brain version of the model introduced and fully described in [37,58] which will describe the activity of each node of the model. Here, we recall the model's equations for one node. The model has three variables, the mean population voltage $h$, the short-term synaptic facilitation $x$, and the short-term synaptic depression $y$:

$$\tau_0 \dot{h} = -(h - T_0) + Jxy(h - T_0)^+ + \sqrt{\tau_0}\sigma\dot{\omega}$$

$$\dot{x} = \frac{X_{eq} - x}{\tau_f} + K(1 - x)(h - T_0)^+ \qquad (2)$$

$$\dot{y} = \frac{1 - y}{\tau_r} - Lxy(h - T_0)^+,$$

where $h^+ = max(h, 0)$ is the population mean firing rate [83,84], $J$ is the average local connectivity parameter [85,86], i.e., within one local population (or node), the influence of the other nodes will be added below in a similar way (see Eq. 3). The parameters $K$ and $L$ describe how the firing rate influences the duration and probability of vesicular release respectively. The time scales $\tau_f$ (for facilitation) and $\tau_r$ (for depression) define the recovery rates of a synapse from the local network activity and $X_{eq}$ is the resting-state value for the facilitation, which can be associated to the presynaptic calcium concentration. $\dot{\omega}$ is an additive Gaussian noise and $\sigma$ its amplitude. Finally, we accounted for AHP with two features: 1) a varying equilibrium state $T_0$ representing hyperpolarization at the end of a burst of neuronal activity 2) two timescales for the medium and slow AHP recovery rates.

The resting membrane potential $T_0$ and the recovery time constant $\tau_0$ of the voltage $h$ are defined piece-wise as follows:

- $\tau_0 = \tau$ and $T_0 = 0$ when $\{y > Y_{AHP}$ and $h \geq H_{AHP}\}$, represents the fluctuations around the resting membrane potential and the bursting dynamics.

- $\tau_0 = \tau_{mAHP}$ and $T_0 = T_{AHP} < 0$ for $\{y < \frac{1}{1 + Lx(h - T_0)}(\iff \dot{y} > 0)$ and $y < Y_h\}$ when the hyperpolarizing currents at the end of the burst become dominant and force the voltage to hyperpolarize (medium AHP period).

- $\tau_0 = \tau_{sAHP}$ and $T_0 = 0$ for $\{y < \frac{1}{1 + Lx(h - T_0)}$ and $(Y_{AHP} < y$ or $h < H_{AHP})\}$, which represents the slow recovery to resting membrane potential (slow AHP period).

The threshold parameters defining the three phases are $Y_h = 0.5$, $Y_{AHP} = 0.85$ and $H_{AHP} = -7.5$.

**Generalization to a whole-brain model.** To generalize this model to a whole-brain version, we just need to add the long-range connections, i.e., the influence of the other nodes of the model, as follows. For node $i$ we now have:

$$\tau_{0,i}\dot{h}_i = -(h_i - T_0) + \sum_{j=1}^{d_opt} J_{ij}x_jy_j(h_j - T_{0,i})^+ + \sqrt{\tau_{0,i}}\sigma\dot{\omega}$$

$$\dot{x}_i = \frac{X_{eq} - x_i}{\tau_f} + K(1 - x_i)(h_i - T_{0,i})^+$$ (3)

$$\dot{y}_i = \frac{Y_{eq} - y_i}{\tau_r} - Lx_iy_i(h - T_{0,i})^+.$$

Similarly as for the Hopf model, the intrinsic timescale $\tau$ is extracted directly from the data using the Spectral Edge Frequency at 50% (SEF50) from the spectrogram of the signal. Similarly as for the Hopf model, we directly fit a connectivity matrix $J_{ij} = G \times C_{ij}$, integrating the global coupling into the fitted connectivity matrix. Note that we have replaced the resting-state value of the depression that was left fixed at 1 in [37,58,84,87] by a parameter $Y_{eq}$ that we will fit to the data to be able to grasp potentially depressed synapses in some patients. The fixed model parameters are taken from [58] and are given in Table 2.

**Balloon-Windkessel model.** To simulate the BOLD signal, we add a Balloon-Windkessel transformation as described in [59,60]. We used the implementation from github.com/dagush/WholeBrain/

**Models implementation and numerical integration.** All models are implemented in Python, with Numba (v0.56.4), a just-in-time compiler, to speed up the computations. The numerical integration for the Hopf and AHP model is done with the Runge-Kutta 4 scheme with a time step of $dt_{sim} = 0.05$s.

## Fitting algorithm

The fitting algorithm is defined in two recurrent steps (see Fig 2A). Both steps are based on maximizing the structural similarity index measure (SSIM) between empirical and simulated connectivity matrices

**SSIM.** The SSIM index is a measure of similarity between two images (or matrices) of the same size [61]. It is computed by comparing common-size windows of corresponding zones of both matrices and then aggregating the results. The formula used for the comparison of two matrices $A$ and $B$ is:

$$SSIM(A, B) = \frac{(2\mu_A\mu_B + c_1)(2\sigma_{AB} + c_2)}{(\mu_A^2 + \mu_B^2 + c_1)(\sigma_A^2 + \sigma_B^2 + c_2)},$$ (4)

where:
$\mu_X$ is the pixel empirical mean of matrix $X \in A, B$ and $\sigma_X$ its empirical variance;
$\sigma_{AB}$ the covariance of the two matrices;
$c_1$ and $c_2$ are two variables introduced to stabilize the division with a weak denominator, we used the classical default values:

**Table 2. Fixed AHP model parameters.**

|  | Parameters | Values |
|---|---|---|
| $K$ | Facilitation rate | 0.037Hz [58,84] |
| $L$ | Depression rate | 0.028Hz [58,84] |
| $\tau_r$ | Facilitation timescale | 2.9s [58] |
| $\tau_f$ | Depression timescale | 0.9s [58] |
| $T$ | voltage resting-state | 0 |

$c_1 = (0.01L)^2$ and $c_2 = (0.03L)^2$ where $L$ is the dynamic range of pixel values in $A$ and $B$, which in our case is $L = 1.0$.

**Node parameters fitting.** First we fit the node parameters, that is $\Theta = a, \sigma$ for the Hopf model and $\Theta = \{\sigma, X_{eq}, Y_{eq}, T_h, 1/\tau_{mAHP}, 1/\tau_{sAHP}\}$ for the AHP model. The goal of this step is to find the choice of node parameters $\Theta$ that will maximize the SSIM between empirical and simulated functional connectivity matrices for each patient. To do so, we keep the connectivity matrix fixed and vary the node parameters $\Theta$ exploring the parameter space.

We explore randomly the parameter space in the manner of a multi-scale grid search, i.e., we first explore a larger range of parameters and at each iteration, we refine the search around the best-found value. In practice, at the first iteration $n_{iter} = 0$, the connectivity matrix $C$ is defined as the empirical functional connectivity matrix, $FC_{emp}$, taken from the data, and for the following iterations, we take $C_{iter} = GEC_{iter-1}$, where $iter$ is the iteration step of the fitting procedure. Here, we stopped at $iter_{max} = 4$ for the Hopf model and $iter_{max} = 6$ for the AHP model, when it was clear that the SSIM stopped increasing with fit iterations.

At each iteration $iter$, we explored $N_{max} = (4 - iter) \times 1000$ (for $iter \leq 3$ and then $N_{max} = 1000$ for the following iterations) values for the node parameters $\Theta$. Since we have a stochastic model we need to run many simulations of the model at each iteration. Thus, for each value of $\Theta$ we ran $N_{sim} = (iter \times 10) + 15$ (for $iter \leq 3$ and then $N_{sim} = 50$ for the following iterations) simulations of duration $dur = 300s$ (same as our recordings) and then averaged them before taking the correlation matrix to obtain the simulated FC $FC_{sim}$. The reason why the number of simulations ran depends on the iteration step $iter$ is to mimic the approach of multi-level grid searches where we first explore a wider range of parameters (i.e., a bigger $N_{max}$) with less precision (i.e., a smaller $N_{sim}$) and then progressively explore fewer parameters (decreasing $N_{max}$) with more precision (increasing $N_{sim}$) while keeping reasonable computation times.

The explored parameter ranges are summarized in Table 3.

**Generative Effective Connectivity fitting** Briefly, the generative effective connectivity matrix (GEC) is a matrix that allows for the reproduction of both the empirical correlation matrix of the data and a forward time-shifted correlation matrix. This is done to grasp the asymmetries in the effective connectivity between the nodes and leads to an asymmetric matrix which can reproduce non-equilibrium dynamics (see, for example, [51,88] for more details about GEC matrices). In practice, we fitted the GEC by keeping the node parameters from the previous step $\Theta$ fixed and adjusting the connectivity matrix to maximize the SSIM between the empirical $FC^{emp}$ vs simulated $FC^{sim}$ matrices, as well as between the shifted matrices $FC^{emp-tau}$ and $FC^{sim-tau}$. The shifted matrices were obtained by shifting in time each of the time series of a small time step $\tau$ and taking its correlation with the unshifted other dimensions. This process is done to grasp the asymmetries in the effective connectivity between the nodes. The GEC was fitted iteratively using a pseudo gradient descent algorithm according to Eq. 5 below, which weighs both the fitting of the direct and shifted FC matrices, as also explained in [51]:

**Table 3. Authorized parameter ranges.**

|  | All | Hopf | AHP model | | | | |
|---|---|---|---|---|---|---|---|
|  | $\sigma$ | a | $X_{eq}$ | $Y_{eq}$ | $T_h$ | $\tau_{mAHP}$ | $\tau_{sAHP}$ |
| Initial value | 5 | -0.02 | 0.088 | 1 | -30 | 0.15s | 5s |
| Range | [0.5,7.5] | [−0.5,0.5] | [0.01,0.21] | [0.1,1] | [−50,−10] | [0.05,0.5]s | [0.75,15]s |

$$GEC_{ij} = GEC_{ij} + \epsilon(FC^{emp} - FC^{sim} + FC^{emp-tau} - FC^{sim-tau}). \tag{5}$$

The learning rate is $\epsilon = 0.01$ and the shift in time $\tau = 3 \times TR = 6s$ [51]. Similarly, as in the node fitting step, we need to run several simulations at each step to account for the stochasticity, here we have $N_{sim} = 5$. For each iteration, this is repeated until the SSIM is larger than 0.95 or for $N_{max} = 1000$ iterations.

## GEC and graph measures

All the graphs are constructed with the *from_numpy_matrix* function of the package *networkx* (v2.8.4). We then split all the graphs between two subgraphs, one containing all the positive edges, and another one containing only the negative edges. For the interclique analysis (Fig 3C) we use the *find_cliques* function to get the maximum clique in which one particular node is involved and take all the other nodes (the ones not contained in the maximal clique) as the other clique, and then computed the average and standard deviation of the connectivity between these two groups of nodes using the *edge_boundary* function, this is repeated for all nodes. The other graph measures (Fig 3D) are directly obtained with built-in *networkx* functions for node centrality and clustering coefficients.

The GEC quantifications (Fig 3E) are computed as follow:

$$sparsity = \frac{n_{small}}{d_{opt}^2}, \tag{6}$$

where $n_{small}$ is the number of values in GEC that are smaller (in absolute value) than 0.015.

$$asymmetry = \frac{|GEC - GEC^T|}{2} \tag{7}$$

The mean connectivity and standard deviation are the mean value of the GEC and its standard deviation (computed with the *numpy* built-in functions).

## Decoding of the GEC cliques and latent dimensions

To understand the connection between the latent dimensions and the natural brain structures, we created surrogate signals in the natural space corresponding to each latent dimension, as follows:

1. We projected all the control subjects' data into the latent space, using the same trained AE as for the rest of the study.
2. From each encoded subject, we created $d_opt = 15$ surrogate signals, by keeping the encoded data on one of the latent dimensions $LD \in [[0, 14]]$ and setting the other ones to zero.
3. We decoded the surrogate signals (with the decoder part of the same trained AE) to project them back into the natural space. At this stage, we obtained 15 surrogate fMRI signal (with 100 time-series – the dimension of the natural space) per control subject.
4. We averaged the decoded fMRI signals over all control subjects. This was done to strengthen the results' robustness, after we verified that, as expected, all decoded signals per latent dimension had a very similar structure, regardless of the dataset (Paris or Liege).

Then we compared the obtained signals for each latent dimension to the 7 resting state networks (RSNs) from [54]. To do so, we computed a FC matrix for each RSN, in the natural space, by keeping the control subjects' data only in the nodes corresponding to one of the RSNs and putting very small amplitude white noise on the other dimensions (generated with $10^{-3} \times numpy.random.rand$). The small noise on the remaining dimensions was added to to avoid the division by zero in the computation of the correlation coefficient for signals that would otherwise have a variance equal to zero.

Finally, we computed the SSIM between the decoded FC matrices for each latent dimension ($LD \in [[0, 14]]$) and the FC matrices obtained for each of the 7 RSNs.

We applied the same methodology for the two cliques of the GEC observed in Fig 3B. This means that instead of keeping the encoded signal only one latent dimension at a time, we kept the signal on all the nodes of one clique and then of the other one. The rest of the analysis process wa kept the same.

### Local node MBBs clustering

**Outcome categorization.**   To analyze the link between the MBB clusters and the patients metadata, we defined five outcome categories for the patients based on the successive CRS-R diagnosis available:

1. Dead;
2. Decreasing: when the evolution of the CRS-R diagnosis went towards less conscious over successive checks but did not lead to death;
3. Stable: when the CRS-R diagnosis did not change over time;
4. Increasing: when the diagnosis ameliorated without leading to consciousness;
5. Conscious or Emerged.

The final outcome available in our dataset is the CRS-R diagnosis after two years unless the patient died or regained consciousness before that.

We refer to outcomes 1 to 3 as *negative* outcomes, and outcomes 4 and 5 as *positive* or *favorable* outcomes.

**UMAPS and HDBSCAN clustering.**   To obtain the clusters we used Uniform Manifold Approximation and Projection for Dimension Reduction (UMAP) [63] on the 6-dimensional (resp. 2-d) vectors of fitted node parameters (node MBBs) from the AHP (resp. Hopf) model, using the UMAP Python package (*umap-learn* v0.5.5), we used the parameters *n_neighbors* = 5 and *min_dist* = 0.1, to project in 2D, the rest are the default parameters.

The clustering was then done on the UMAP projections using the *HDBScan* clustering algorithm [64] implemented for Python in the *HDBSCAN* package (*hdbscan* v0.8.33). We used parameters *min_samples* = 3 and *min_cluster_size* = 3, and the rest take default values.

## Supporting information

**S1 File. Supplementary figures for the Hopf model.**
PDF

## Author contributions

**Conceptualization:** Lou Zonca, Jacobo Diego Sitt, Gustavo Deco.

**Data curation:** Anira Escrichs, Dragana Manasova, Jitka Annen, Olivia Gosseries, Steven Laureys.

**Formal analysis:** Lou Zonca.

**Funding acquisition:** Olivia Gosseries, Jacobo Diego Sitt, Gustavo Deco.

**Investigation:** Lou Zonca.

**Methodology:** Lou Zonca, Yonathan Sanz-Perl.

**Project administration:** Lou Zonca, Jacobo Diego Sitt, Gustavo Deco.

**Software:** Lou Zonca, Gustavo Patow.

**Supervision:** Jacobo Diego Sitt, Gustavo Deco.

**Validation:** Lou Zonca.

**Visualization:** Lou Zonca.

**Writing – original draft:** Lou Zonca.

**Writing – review & editing:** Lou Zonca, Anira Escrichs, Gustavo Patow, Dragana Manasova, Yonathan Sanz-Perl, Jitka Annen, Olivia Gosseries, Steven Laureys, Jacobo Diego Sitt, Gustavo Deco.

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
