## [Decision Letter · Decision Letter 0]

11 Mar 2025

PONE-D-24-54958Modeling disorders of consciousness at the patient level reveals the network's influence on the diagnosis vs the local node parameters role in prognosisPLOS ONE

Dear Dr. Zonca,

Thank you for submitting your manuscript to PLOS ONE. After careful consideration, we feel that it has merit but does not fully meet PLOS ONE’s publication criteria as it currently stands. Therefore, we invite you to submit a revised version of the manuscript that addresses the points raised during the review process.

We look forward to receiving your revised manuscript.

Kind regards,

Marie-Constance Corsi

Academic Editor

PLOS ONE

Journal Requirements:

 “LZ was supported by the FLAG-ERA JTC2021 project ModelDXConsciousness (Human Brain Project Partnering Project) Grant PCI2021-122019-2A funded by MICIU/AEI /10.13039/501100011033 and by the European Union NextGenerationEU/PRTR. AE was supported by the project eBRAIN-Health—Actionable Multilevel Health Data (id 101058516), funded by EU Horizon Europe and by the Grant PID2022-136216NBI00, funded by MICIU/AEI/10.13039/501100011033, and “ERDF A way of making Europe”, ERDF, EU. YSP was supported by the project NEurological MEchanismS of Injury, and the project Sleep-like cellular dynamics (NEMESIS) (ref. 101071900) funded by the EU ERC Synergy Horizon Europe. GD was supported by the project NEurological MEchanismS of Injury, and the project Sleep-like cellular dynamics (NEMESIS) (ref. 101071900) funded by the EU ERC Synergy Horizon Europe and and by the Grant PID2022-136216NB-I00, funded by MICIU/AEI/10.13039/501100011033, and “ERDF A way of making Europe”, ERDF, EU.

This research was partially funded by Grant PID2021-122136OB-C22 funded by MICIU/AEI/10.13039/501100011033 and by ERDF A way of making Europe for GP.

DM received individual funding from Ecole Doctorale Frontières de l’Innovation en Recherche et Education–Programme Bettencourt.

DM and JDS were supported by funding from the EU ERAPerMed Joint Translational Call for Proposals for “Personalised Medicine: Multidisciplinary research towards implementation” (ERA PerMed JTC2019). JDS was also supported by the FLAG-ERA JTC2021 project ModelDXConsciousness (Human Brain Project Partnering Project).

The study was further supported by the University and University Hospital of Liège, the BIAL Foundation, the Belgian National Funds for Scientific Research (FRS-FNRS), the FNRS PDR project (T.0134.21), the FLAG-ERA JTC2021 project ModelDXConsciousness (Human Brain Project Partnering Project) and FLAG-ERA JTC 2023 - HBP - Basic and Applied Research, project BrainAct, JTC the fund Generet, the King Baudouin Foundation, the Funds Chantal Schaeck Yolande, the Télévie Foundation, the European Space Agency (ESA) and the Belgian Federal Science Policy Office (BELSPO) in the framework of the PRODEX Programme, the Public Utility Foundation ‘Université Européenne du Travail’, “Fondazione Europea di Ricerca Biomedica”, the BIAL Foundation, the Mind Science Foundation, the European Commission, the Fondation Leon Fredericq, the Mind-Care foundation, the National Natural Science Foundation of China (Joint Research Project 81471100) and the European Foundation of Biomedical Research FERB Onlus, the Horizon 2020 MSCA – Research and Innovation Staff Exchange DoC-Box project (HORIZON-MSCA-2022-SE-01-01; 101131344). OG is Research Associate at FRS-FNRS. JA is postdoctoral fellow funded (1265522N) by the Fund for Scientific Research-Flanders (FWO).”

Reviewers' comments:

Reviewer's Responses to Questions

**Comments to the Author**

1. Is the manuscript technically sound, and do the data support the conclusions?

Reviewer #1: Yes

2. Has the statistical analysis been performed appropriately and rigorously? 

Reviewer #1: Yes

3. Have the authors made all data underlying the findings in their manuscript fully available?

Reviewer #1: Yes

4. Is the manuscript presented in an intelligible fashion and written in standard English?

Reviewer #1: Yes

5. Review Comments to the Author

Reviewer #1: The manuscript presents a framework for modeling disorders of consciousness (DoC) at the individual patient level by combining dimensionality reduction techniques and biologically inspired whole-brain modeling. The main finding—that connectivity matrices and local parameters carry complementary information regarding the patient's diagnosis and prognosis—is highly relevant.

The strengths of this study are manifold. The latent space framework, employed in recent publications from the same team, simplifies data complexity while retaining essential diagnostic features, demonstrating robustness across different datasets, including the two DoC datasets analyzed in the present manuscript.

Connectivity-based brain models are then simulated to reproduce the latent dynamics in silico. The authors employ a recently introduced astrocyte-inspired model (AHP) alongside the more established Hopf model. Integrating biologically meaningful parameters, such as those derived from the AHP model, is another major contribution, as it offers a pathway to understanding underlying brain mechanisms disrupted in DoC, as the authors underscore in the discussion. The authors have made the pipeline available, which helps transparency.

The potential applications of model-based biomarkers (e.g., connectivity and local node parameters), both in clinical decision-making and personalized treatment, highlight the broad interest of this work.

Despite its strengths, the manuscript needs significant revisions to enhance clarity and methodological robustness.

1) One major issue pertains to the connectivity analysis and its application across datasets. It is unclear whether the authors applied the analysis to both the Paris and Liege datasets or only to one of them. If (as I understand) both datasets are used for the results in Figure 3, mixing CNT, MCS, and UWS groups from two different datasets could result in spurious classification, as the datasets are acquired at different resolutions using different pipelines (despite the latent space method, which might partly suppress these differences). Confirming the effectiveness of the classification separately on the two datasets would strengthen the validity of the results.

2) Another area that needs further clarification is the interpretation of connectivity results. The manuscript identifies positive and negative connections within the generative effective connectivity (GEC) matrices but stops short of exploring their interpretation. A discussion on the meaning of positive inter-clique links and negative intra-clique links would enrich the study. For instance, mapping the latent modes back to standard resting state networks (as in Sanz-Perl et al Network Neuroscience 2025) could provide neuroscientific interpretability, helping to connect the findings (about the modular interaction of two cliques) with existing functional network frameworks.

3) The clustering of patients into groups with different outcomes is intriguing but requires a more detailed quantification. For example, the interpretation of Fig.4C reports that “clusters 2 and 3 contain mainly patients with negative outcomes, cluster 1 contains a majority of emerged patients, and clusters 0 and 4 contain more mixed outcomes”. However, there are no numbers displayed to assess if the percentages support the analysis by visual inspection.

4) The manuscript does not provide a rationale for the choice of fixing the global coupling (G=1). Since G is generally regarded as a crucial parameter that influences global network dynamics, wouldn’t it be beneficial to include it in the iterative fitting process alongside local parameters? It would be helpful if the authors provided a justification for why G is held constant.

5) The Methods section would benefit from greater transparency and detail. Key metrics, such as the structural similarity index (SSIM), are referenced but not explicitly defined. Including a brief description of the SSIM formula would make the study self-contained. Similarly, the procedures for fitting model parameters and the rationale behind the specific GEC formula are not described with sufficient clarity. While it is reasonable to reference prior work for context, the paper would be easier to follow if these core methods were explicitly presented.

6) Beyond methodological issues, the manuscript’s writing could be improved to enhance readability. The use of inconsistent tenses, punctuation errors, and occasional typos (e.g., “…simulations is Nmax is reached”, or “. Where the shifted matrices are obtained..”, or “using the edge_boundary function”) detract from the clarity of the scientific content. I recommend using tools like Grammarly to identify and correct these issues. Additionally, some sentences are overly verbose or unclear, such as “Moreover, analysis of the clusters’ separability also revealed which parameter, or group of parameters, were leading to belong to which cluster” which could be rephrased more concisely. A thorough revision for grammar, style (e.g., reducing the excessive use of the Saxon genitive 's), and logical flow would make the paper more accessible to the reader and better communicate its scientific value.

7) A revised title might help improve the paper's visibility. The current title, “Modeling disorders of consciousness at the patient level reveals the network's influence on the diagnosis vs the local node parameters role in prognosis”, feels a bit lengthy. Additionally, using “vs” is not ideal. You might consider something like: “Personalized models of Disorders of Consciousness reveal complementary roles of connectivity and local parameters in diagnosis and prognosis”.

8) The figures, while generally informative, also require some adjustments. Missing labels, such as for the clusters’ IDs, or the absent color bar in functional connectivity figures, should be added to ensure clarity. In Figure 4A, the yellow cluster is nearly invisible, making it difficult to interpret the results. Adjusting the color palette or improving the visual contrast would address this issue. Figure S2 title should be “GEC-based biomarkers for the Hopf model:” and panel C should be adjusted in the caption “bf C.”.

9) Notice that PONE guidelines specify that “If there are restrictions on publicly sharing data […] those must be specified.”

In conclusion, this study presents a promising framework for analyzing DoC patients, combining innovative methodologies with biologically grounded insights. However, substantial revisions are required to address the issues outlined above. Expanding the connectivity analysis to both datasets or justifying their combined analysis would enhance the manuscript's robustness, while clearer interpretations and improved methodology transparency will help realize the full potential of this promising framework. I hope that these comments will assist in improving the manuscript.

Sincerely,

Giovanni Rabuffo

6. PLOS authors have the option to publish the peer review history of their article (what does this mean?). If published, this will include your full peer review and any attached files.

Reviewer #1: **Yes: **Giovanni Rabuffo

---

## [Author Response · Author response to Decision Letter 1]

9 May 2025

We have attached our response as a PDF in the re submitted files.

---

## [Decision Letter · Decision Letter 1]

15 Jun 2025

PONE-D-24-54958R1Personalized models of Disorders of Consciousness reveal complementary roles of connectivity and local parameters in diagnosis and prognosisPLOS ONE

Dear Dr. Zonca,

Thank you for submitting your manuscript to PLOS ONE. After careful consideration, we feel that it has merit but does not fully meet PLOS ONE’s publication criteria as it currently stands. Therefore, we invite you to submit a revised version of the manuscript that addresses the points raised during the review process.

We look forward to receiving your revised manuscript.

Kind regards,

Marie-Constance Corsi

Academic Editor

PLOS ONE

Journal Requirements:

Additional Editor Comments (if provided):

Reviewers' comments:

Reviewer's Responses to Questions

**Comments to the Author**

1. If the authors have adequately addressed your comments raised in a previous round of review and you feel that this manuscript is now acceptable for publication, you may indicate that here to bypass the “Comments to the Author” section, enter your conflict of interest statement in the “Confidential to Editor” section, and submit your "Accept" recommendation.

Reviewer #1: All comments have been addressed

2. Is the manuscript technically sound, and do the data support the conclusions?

Reviewer #1: Yes

3. Has the statistical analysis been performed appropriately and rigorously? 

Reviewer #1: Yes

4. Have the authors made all data underlying the findings in their manuscript fully available?

Reviewer #1: Yes

5. Is the manuscript presented in an intelligible fashion and written in standard English?

Reviewer #1: Yes

6. Review Comments to the Author

Reviewer #1: (No Response)

7. PLOS authors have the option to publish the peer review history of their article (what does this mean?). If published, this will include your full peer review and any attached files.

Reviewer #1: **Yes: **Giovanni Rabuffo

---

## [Author Response · Author response to Decision Letter 2]

25 Jun 2025

All our replies are attached in the response to reviewers letters.

---

## [Editor Report · Decision Letter 2]

30 Jun 2025

Personalized models of Disorders of Consciousness reveal complementary roles of connectivity and local parameters in diagnosis and prognosis

PONE-D-24-54958R2

Dear Dr. Zonca,

We’re pleased to inform you that your manuscript has been judged scientifically suitable for publication and will be formally accepted for publication once it meets all outstanding technical requirements.

Kind regards,

Marie-Constance Corsi

Academic Editor

PLOS ONE
---

## [Editor Report · Acceptance letter]

PONE-D-24-54958R2

PLOS ONE

Dear Dr. Zonca,

I'm pleased to inform you that your manuscript has been deemed suitable for publication in PLOS ONE. Congratulations! Your manuscript is now being handed over to our production team.

Kind regards,

on behalf of

Dr. Marie-Constance Corsi

Academic Editor

PLOS ONE